# From Oxytocin to Compassion: The Saliency of Distress

**DOI:** 10.3390/biology12020183

**Published:** 2023-01-25

**Authors:** Barbora Kucerova, Nava Levit-Binnun, Ilanit Gordon, Yulia Golland

**Affiliations:** 1Psychology Department, Faculty of Social Studies, Masaryk University, 602 00 Brno, Czech Republic; 2Baruch Ivcher School of Psychology, Reichman University, Herzliya 4610101, Israel; 3Department of Psychology, Bar-Ilan University, Ramat Gan 5290002, Israel

**Keywords:** oxytocin, compassion, distress cues, saliency, individual differences

## Abstract

**Simple Summary:**

Compassion is a warm response of care and concern for those who are suffering, which drives individuals to devote their resources for the sake of others. A prominent evolutionary framework grounds compassion in the neurobiology of the caregiving system - the inborn motivation to support close others in times of need. The current review paper investigates the neurobiological mechanisms of compassion, and in particular, the neuropeptide oxytocin which is known to play a significant role in caregiving. Integrating the state-of-the-art literature on oxytocin with the current understanding of compassion, we suggest that oxytocin regulates the necessary precursor of compassion, i.e., the saliency of pain and distress cues. In according to this view, social cues are more likely to be noticed and responded to in an oxytocin-enriched environment. This framework focuses on the saliency processes in the context of social suffering. It sheds light on two central aspects of compassion. First, it highlights the less investigated initial stages of compassion, i.e. sensitivity and attentiveness to pain and distress. Second, it opens the door to the vast variability of social responses when facing suffering others, by demonstrating that emotions, motivations and behaviors elicited by such sensitivity to pain are crucially dependent on context and personality.

**Abstract:**

Compassion is a warm response of care and concern for those who are suffering, which drives individuals to devote their resources for the sake of others. A prominent neuroevolutionary framework grounds compassion in the neurobiology of the mammalian caregiving system. Accordingly, it has been suggested that the oxytocinergic system, which plays a central role in parental caregiving and bonding, provides the neurobiological foundation for compassion towards strangers. Yet, the specific role of oxytocin in compassion is far from clear. The current paper aims to target this gap and offer a theoretical framework that integrates the state-of-the-art literature on oxytocin with research on compassion. We suggest that oxytocin mediates compassion by enhancing the saliency of cues of pain and distress and discuss the plausible underlying neurobiological substrates. We further demonstrate how the proposed framework can account for individual differences in compassion, focusing on the effects of attachment on caregiving and support. The proposed framework integrates the current scientific understanding of oxytocin function with compassion-related processes. It thus highlights the largely ignored attentional processes in compassion and taps into the vast variability of responses in social contexts involving pain and suffering.

## 1. Introduction

Compassion is a warm response of care and willingness to alleviate suffering, which may have evolved in humans to increase survival [1]. Indeed, from prehistoric to modern times, sustained care for those in need has been an integral part of human society [2]. Compassion has been frequently described as the motivational force of empathy, manifested by a desire to alleviate others’ pain [1,3,4]. Centrally, it predicts actual pro-social behavior and help [5,6]. Understanding the mechanisms of compassion is highly valuable, as compassion cultivation has been widely linked to multiple beneficial effects for individuals and society [7,8,9].

A crucial question in compassion research concerns the neurobiological mechanisms driving these other-oriented motivations, which can arise despite significant self-costs. Multiple voices in the literature have suggested that the mammalian caregiving system, evolutionarily designed to ensure caring for offspring, plays a pivotal role in the compassionate motivation to attend to and care for less fortunate members of a group [10,11,12,13]. Just as parental instincts foster offspring survival, compassion promotes the welfare of society and its members [1]. In particular, it has been suggested that the evolutionary ancient oxytocinergic system, which plays a central role in parental caregiving and bonding [14], provides the neurobiological foundation for compassion towards kin and non-kin strangers [11,15,16].

These theoretical claims, which assign a pivotal role to oxytocin in the context of compassion, are supported by substantial empirical findings linking oxytocin with various indexes of care and empathy. For instance, a comprehensive series of studies employing acute intranasal oxytocin treatment have shown that exogenous oxytocin significantly increased emotional empathy (i.e., feeling the same as the character) and compassion (i.e., feeling for the character) towards pictorial characters in emotionally charged situations [17,18,19,20]. Along similar lines, oxytocin increase compassion when listening to recordings of distressed protagonists [21,22], as well as towards crime victims in a set of legal vignettes [23]. Research has also reported increases in endogenous oxytocin following films that elicit compassion [24,25,26] and after a session of compassion cultivation [27]. Finally, genetic studies have found consistent associations between genetic variations in the oxytocin receptor gene and a trait-level pre-disposition to pro-sociality and sensitivity to others’ suffering [28,29,30,31,32].

Despite this body of evidence and common theoretical claims, the specific functional role of oxytocin in compassion remains surprisingly unclear. A widely accepted framework suggests that oxytocin exerts its effects by optimally orchestrating reward- and stress-related systems and their interconnections [33,34,35,36]. Accordingly, in the context of suffering, a release of oxytocin triggers pro-social approach motivation, driving individuals towards others in distress while attenuating the sense of threat and anxiety [11,15]. However, the univocal anxiolytic and pro-social effects of oxytocin are currently under debate. Multiple studies demonstrate complex and contradictory results [37,38,39,40,41,42,43,44,45], as well as a lack of replication [46,47]. Contemporary models of oxytocin suggest a more complex functional profile in which oxytocin shapes the saliency of social (and non-social) cues in a context-dependent manner [38,48]. Furthermore, the researchers suggest that these saliency processes are not bound to social stimuli but rather reflect general gating and adaptation mechanisms [49,50]. However, integrating these saliency processes into models of compassion has not yet been attempted.

In the current work, we aim to delineate the potential role of oxytocin in eliciting compassionate responses towards distressed strangers while building upon the tremendously rich and evolving science regarding oxytocin. We first present the widely agreed-upon contemporary framework of oxytocin’s function, which highlights its pivotal role in shaping the saliency of social cues in a context-dependent manner [38]. We then demonstrate how oxytocin-driven saliency of social cues may underlie compassionate responses and suggest that these processes are grounded in the mesolimbic reward circuitry. Finally, we explore how saliency processing may mediate the vast individual differences in compassion and pro-sociality while bringing evidence from the perspective of attachment research.

## 2. The Social Saliency Hypothesis

The researchers increasingly acknowledge that the effects of oxytocin are not uniform and are more complex than previously believed [51]. For instance, despite its known anxiolytic effects, oxytocin can also lead to increased fear and anxiety. Accordingly, oxytocin administration increased anxiety regarding unpredictable threats [52] and potentiated stress responses to negative social stimuli [53]. In addition, in contexts involving competition or out-group members, the administration of oxytocin decreased cooperation [42] and increased envy [45] as well as aggression [41]. In the non-social domain of chronic pain, while animal studies suggest that oxytocin may have anxiolytic functions, results from human studies are mixed and inconsistent [54]. The vast variability in the effects of exogenous oxytocin has led the field to acknowledge its mechanisms’ complexity and search for new models [51]. To reconcile the diverse empirical findings, the social saliency hypothesis (SSH) was formulated [38]. The SSH suggests that, rather than inducing pro-social effects, oxytocin plays a vital role in regulating the salience of social cues by modulating overt and covert attentional shifts. Accordingly, in an oxytocin-enriched state or environment, social cues have a higher potential to be noticed, identified, and responded to.

Overall, the SSH has received substantial empirical support. Central to the current study, research on social attention has consistently demonstrated that oxytocin increases attention towards social but not non-social stimuli in humans and monkeys [20,55,56,57,58,59,60]. Furthermore, oxytocin can improve the detection and recognition of emotional signals by modulating low-level attentional processes, specifically by increasing the time of gaze and number of saccades towards the eye region of facial stimuli [61,62,63,64], even in young children [65] and, in particular, for faces expressing fear [66,67]. However, other studies have failed to find these effects [68,69,70]. A recent study directly testing the SSH’s predictions examined attentional and emotional responses to (social) mortality threat cues in a visual dot-prob task [71]. The experiment showed that the intranasal oxytocin enhanced participants’ attentiveness to mortality threat cues compared with a placebo, which provided direct support for the SSH. Therefore, in contrast to the predictions of the pro-social/anxiolytic models, participants in the oxytocin group reported higher anxiety levels.

To summarize, previous research on social attention and recognition of social-emotional signals has shown that intranasal oxytocin enhanced attention to social stimuli and affected the saliency of critical features of these stimuli. This impressive body of research highlights saliency signaling as a central role of oxytocin, which may or may not lead to pro-sociality.

## 3. From Social Saliency to Compassion

While there are many definitions of compassion, all of them seem to agree that compassion encapsulates attentiveness and sensitivity to social signals of distress [3,72]. Signals of distress, such as crying, frowning, and sobbing, are essential for eliciting care and help in close attachment systems [73,74,75]. Similarly, the ability to identify distress cues is a significant antecedent of compassionate responses to strangers [76,77,78]. In support of this view, it has been shown that compassion training increased attention to images depicting human suffering compared with neutral images [79].

Integrating the SSH with the significance of distress cues for caring, we suggest that oxytocin may modulate the saliency of distress cues which, in turn, can significantly shape compassionate responses. A recent pharmacological eye-tracking study showed that oxytocin administration enhanced visual attention to emotionally salient regions, i.e., faces. Notably, increased attention to the face region in negative scenes was correlated with increased empathy rates for those pictured [20]. Other studies have shown that oxytocin improved emotion recognition according to visual cues [80,81,82], in particular in the case of fear [83], which is believed to elicit caring responses and motivations [74]. Finally, oxytocin administration enhanced empathic accuracy for less socially equipped individuals [84].

To summarize, we suggest that oxytocin modulates the attentional, perceptual, and motivational aspects of distress processing. Accordingly, oxytocin is not a mechanism of compassion per se, in the sense that increases in oxytocin causally lead to increases in compassion. Instead, oxytocin regulates a key saliency process that has the potential to impact compassion for others, as well as a variety of other social behaviors, depending on the context and individual differences.

## 4. A Putative Neurobiological Mechanism of Social Saliency Evaluation

The mesolimbic dopaminergic reward system is the brain’s central motivation system. It signals the significance of stimuli, assigns motivational value, and selects appropriate action, such as approach or avoidance. The dopaminergic reward circuitry is believed to encode "incentive salience," i.e., the attention-grabbing and motivational features of rewards and their learned cues [85,86]. Remarkably, several lines of evidence suggest that the effects of oxytocin on saliency may be mediated via its interaction with dopaminergic pathways [38,87]. Oxytocin neurons originate in the paraventricular nucleus of the hypothalamus (PVN) and send projections to extrahypothalamic regions, which include the two critical regions of the dopaminergic reward circuitry, i.e., nucleus accumbens and ventral tegmental area (VTA). These regions are also known to contain a large number of oxytocin receptors, stimulation of which has directly impacted both dopaminergic activity and motivated behavior patterns in social contexts [88]. For example, intranasal oxytocin significantly increased activation of the VTA in response to social cues [89]. A series of elegant animal studies have shown that pharmacological activation of this oxytocinergic pathway to the brain’s reward system increased social approach behaviors, while its inhibition reduced them [90,91,92,93]. In addition, optogenetic activation of oxytocin input into the VTA increased social interactions with a stranger [92]. Overall, this research suggests that oxytocin directly modulates the reward value of social contact, increasing the saliency of social signals and instigating dopamine-infused motivation to approach and interact with conspecifics in neutral as well as in stressful situations [87,92,94,95].

In line with this model, neuroimaging research has demonstrated that activation of the brain’s reward circuitry is directly linked to compassionate responses to suffering [96,97], as well as altruism and helping [98,99]. In one of the most persuasive demonstrations, moment-to-moment activity in the reward circuitry, particularly in the nucleus accumbens, was shown to predict a momentary shift in caring feelings toward suffering protagonists of autobiographical stories. This activity also predicted monetary donations [96]. Accordingly, the dopaminergic reward system is commonly accepted as the critical neural mechanism underlying the pro-social approach motivation to others who are suffering [4,100].

To summarize, several converging lines of evidence have suggested that neural modulation via oxytocin of the dopaminergic reward-related neural circuits may play a pivotal role in driving compassionate responses to suffering. Centrally, this neurobiological model combines the attentional-perceptual aspects of the SSH described in the previous section and the pro-social motivation aspects of earlier models of oxytocin [11,15,36] by tying salient social cues with rewarding value. When applied to the field of compassion, this model suggests that the effects of oxytocin on compassion are grounded in the motivational state to attend to others, triggered by distress cues and instantiated by dopamine-associated reinforcement pathways [101,102].

## 5. Individual Differences in Compassion

Humans are not routinely compassionate when facing others in distress. The willingness and motivation to care for others are fragile and highly dependent on individual characteristics, available resources, and social context [4,78]. Research has indicated that the path from recognizing a need for help to compassionate response can be moderated by individual differences, including general emotionality, emotion regulation, psychopathy, attachment style, and gender [76,103].

The diversity of social behaviors in empathic contexts resonates well with the SSH, which suggests that the effects of oxytocin-driven saliency of social cues vary as a function of context as well as individual differences of the perceivers [38,48,104]. For instance, in competitive or threatening contexts, oxytocin appears to increase the salience of threat signals and to intensify defensive and anti-social behaviors, such as competitiveness, aggressiveness, and envy [41,42,45]. In cooperative, affiliative contexts, oxytocin seems to increase the saliency of safe cues, facilitating approach, and collaborative pro-social behaviors [105]. In addition, a substantial portion of the reported effects of oxytocin is limited to particular subgroups, with gender, personality traits, and attachment style being the most frequently reported moderators [48,105].

Combining the SSH framework with evidence of individual differences in compassion [4,106], we suggest that oxytocin may mediate individual differences in compassionate behaviors by interacting with personality and contextual factors at different processing stages (see Figure 1). First, individual differences may affect the basal level of oxytocin [107] or the degree of oxytocin released in response to social cues [108]. Thus, the saliency of such cues is either increased or decreased and so is the perception of need. In addition, individual and contextual differences may shape the response to salient social cues. For example, for perceivers with emotion regulation difficulties, increasing the salience of distress signals could elicit avoidance strategies and behaviors [49,109]. Individuals with a high sense of social power have a lower propensity to care for others in distress, even though they have equally detected the need for help [110]. Finally, highly empathetic individuals are more prone to physiological contagion than those with lower dispositional empathy [111].

In the above section, we showed how oxytocin-driven processes might underlie the variability of behaviors in social contexts, increasing care and compassion if certain conditions are met but inhibiting it if other conditions occur. In the next section, we demonstrate how the saliency processes interact with the vital personality characteristic in the context of compassion and care, i.e., attachment patterns. 

## 6. Through the Lens of Attachment

One of the most comprehensive frameworks for individual differences in caring and compassion is the attachment theory, which concerns the effects of early life close relationships on patterns of thought and behavior in subsequent relationships [109,112,113]. According to this framework, individuals’ attachment and caregiving systems are closely intertwined [114]. Specifically, the development of an optimal attachment system and the resulting sense of security facilitates the functioning of the caregiving system that governs pro-social behavior and compassionate acts of helping others in need [109,113,114,115]. Indeed, attachment security developed in early life stages predict higher levels of compassion and caregiving across the lifespan, both in close relationships and towards strangers [116,117,118,119].

Significantly, attachment security does not simply boost the propensity for social engagement. In resonance with the SSH, attachment security facilitates sensitivity and attunement to social cues, potentially via the modulation of social signals saliency through the oxytocin pathways [14,34,36,120]. For example, parent-infant social synchrony–i.e., the synchronization of parental behavior and their infant’s social signals–has been consistently linked to attachment security and higher oxytocin levels in parents [14,34,112,121,122].

Seminal studies on the provision of support in romantic relationships have provided an exquisite example of how the saliency of distress cues on behalf of the support seeker can elicit different experiences and behaviors from the support provider as a function of their attachment patterns [123]. For partners with secure attachment, distress expressivity, i.e., higher saliency of need, predicted compassionate feelings and behaviors [73,123,124]. However, these close links between cues of distress and supportive behaviors are disrupted in insecure caregivers [123]. Thus, insecure-anxious caregivers tend to be out of sync with their partner’s needs; they are over-involved and show high levels of empathy regardless of their partner’s level of distress. Furthermore, they show elevated adverse emotional reactions to their partner’s distress, which causes them to become more self-focused, angry, and frustrated [124]. Similar impediments of sensitive support behaviors due to attachment insecurities have also been demonstrated by studies focusing on compassion towards strangers in distress [125,126,127,128].

Overall, these studies demonstrate that individuals with secure attachment are more sensitive to the needs of others and are better able to modulate their attentional, emotional, and motivational processes in a way contingent on their partners’ expressions of distress [123]. In contrast, individuals with insecure attachment lack the responsiveness to signals of distress either due to lower saliency of distress cues or due to providing an inadequate response to such cues.

It is plausible that the above-described effects of attachment on compassion are mediated via oxytocinergic pathways as attachment and oxytocin exhibit intricate interactions. Both human and rodent studies have demonstrated that oxytocin plays a central role in forming and manifesting attachment and caregiving systems [36,87,129,130,131]. Convergent translational and developmental research has indicated that individuals’ unique early life experiences, i.e., attachment history, shape the function of their oxytocin system [130,132], which may explain later social variability [104,133]. For instance, mothers with secure attachment exhibited greater activation of the oxytocin-associated hypothalamus/pituitary region when viewing their own infant’s smiling and crying faces. Peripheral oxytocin response to infant contact was also significantly higher in secure mothers and positively correlated with brain activation in these regions [108]. Finally, in non-parental social contexts, oxytocin administration has been found to enhance a sense of secure attachment in adults [134]. In contrast, the interaction of oxytocin genotype and attachment might explain social variability [135].

En masse, this evidence suggests that attachment security is linked with basal oxytocin levels and the release of oxytocin in response to social cues. Accordingly, attenuated saliency of social distress cues may mediate the the disrupted support provision in individuals with attachment insecurities. In support, insecurely attached individuals were shown to benefit more from the intranasal administration of oxytocin in social contexts involving trust, intimacy, and cooperation [42,136,137].

## 7. Final Notes and Future Directions

Introducing saliency processes into the functional model of oxytocin has allowed the integration of vast variability of oxytocin effects under one theoretical umbrella. This review has aimed to extend this explanatory model to include the context of compassion. The resulting integrative framework offers two straightforward advances for understanding compassion mechanisms. First, it provides a neurobiological account of the critical component of compassion, namely, sensitivity to cues of suffering [7]. Second, it specifies the mechanisms underlying individual differences in compassion [113]. Although we reviewed an expansive body of literature on oxytocin, only a tiny portion directly focused on its effects in the context of social suffering. We hope the current paper will bring attention to this gap and inspire further research related to the function of oxytocin in encounters with distress and pain. Methodological approaches that tap into the saliency of perceptual cues may inform future compassion research, as we will briefly discuss in the following section.

The saliency model of oxytocin highlights the less investigated attentional processes in compassion. While an essential part of compassion is to be "attentive to the distress and needs of another" [3], little is known about these initial stages that instigate pro-social responses, feelings, and attitudes. One of the most direct and abundant pieces of evidence for the SSH comes from social orientation studies, which demonstrated that exogenous oxytocin modifies low-level attentional processes as manifested by eye-gaze behaviors and emotion recognition in static and dynamic faces, e.g., [67]. Incorporating this methodology in studying the effects of oxytocin on processing compassion-relevant cues, such as facial expressions of sadness and pain or scenes involving distress, can allow for direct investigation of attentional saliency in the context of social suffering, e.g., [79]. In addition, dyadic approaches, derived from relationship research [124] and naturalistic neuroscience [138,139,140], can open the gate to study the saliency of distress in real-life conditions by assessing correspondence or synchrony between the distress cues of one individual and the responses and feelings of another individual. Thus, in our previous work, we showed that participants, who focused on care towards a target in distress, were able to track the dynamics of the target’s unfolding distress continuously. We also found that their physiological responses were de-coupled from these distress fluctuations, suggesting that compassion contains attentional but not body attunement to suffering [139]. Finally, elucidating attentional processes in compassion can inform clinical interventions [7] and shed additional light on how attention-focused mental practices, such as mindfulness, can boost caring and compassion [141].

Integration of oxytocin-driven saliency aspects in compassion-related processes highlights the variable response repertoire in social distress. Caring for others is a costly behavior and, therefore, is generally more likely to be focused on kin and allies [142]. Thus, compassion is not trivial when we face strangers in distress. Furthermore, the context-sensitive caregiving system is attuned to those we like, trust and perceive to be similar [15,103]. Hence, the integrative model presented in this manuscript suggests that contextual and individual factors may shape the saliency of distress cues via the oxytocin pathways. Such saliency regulation can occur in different stages and conditions. First, it can be lower in some individuals due to differences in the basal level of oxytocin and/or personality characteristics. Correspondingly, it has been shown that the administration of oxytocin can enhance pro-social attitudes for people with insecure attachment [42,136,137] or for individuals with greater difficulty in identifying emotions [143]. In addition, attenuation or enhancement of saliency of distress can stem from contextual features (e.g., competition for resources vs. therapeutic goals) or characteristics of the target individual (e.g., an out-group member vs. a child). For example, oxytocin was found to reduce the saliency of the pain of out-group members [144]. Finally, the saliency of suffering may lead to anti-social results, thereby upregulating personal distress and avoidance in individuals with insecure attachment histories or difficulties with emotion regulation [113].

The effects of context and personality on compassion, presented in the previous section, lead to an intriguing conclusion: the dependency of compassion on the neurobiology of mammalian caregiving grounds the human pro-social system in habitual responses that reactivate existing personality patterns and learned associations when facing suffering others [145]. Therefore, reliance on spontaneous, bottom-up compassion is prone to multiple biases. Accordingly, researchers suggest cultivating compassion through cognitive insight and intention [78]. Similarly, a few voices have proposed that compassion encompasses high-level cognitive processes, manifesting concepts of interconnectedness and common humanity [78,146], as well as higher-order awareness of our habitual responses [3]. We join these voices and speculate that overriding the limits of kin-selective oxytocin-driven care can be achieved by higher-order conceptual processes and awareness of our habitual sensitivities.

## Figures and Tables

**Figure 1 biology-12-00183-f001:**
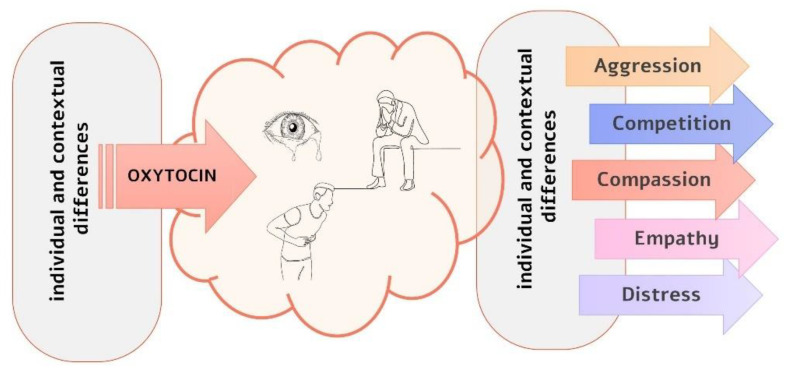
Oxytocin can shape compassion at different processing stages. First, some personality characteristics appear to be related to low basal levels of oxytocin. In addition, similar distress cues may elicit different phasic responses and, consequently, different saliency levels as a function of the personal history or contextual factors in which these cues appear. Second, salient distress cues may elicit different motivations, cognitive processes, and emotional responses in different individuals and contexts.

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
