# Peer review of "From Oxytocin to Compassion: The Saliency of Distress"

_biology, 2023, doi:10.3390/biology12020183_

Round 1

Reviewer 1 Report

An interesting perspective on oxytocin vs. compassion. Or better said, from oxytocin to compassion. I really enjoyed the fresh and original perspective touched by the authors here.  I also believe the readers will be interested in this topic. With oxytocin being a growing trend now in science.  No suggestions really. I think it can be accepted as it is.

Author Response

We thank the Reviewer for the positive feedback on our manuscript

Reviewer 2 Report

The manuscript of  Barbora Kucerova et al. entitled  From oxytocin to compassion: The saliency of distress” aims to review and summarize the physiological roles of oxytocin based on some new aspects. However, the quality of presentation can be improved.

1. Affiliation(s) of authors should be indicated in the manuscript.

2. It should be defined more clearly, what „saliency-related models of oxytocin” does mean.

3. More details about role of oxytocin in the mesolimbic dopaminergic system could be included (source of oxytocin, relevant brain areas, behavioural effects, possible mechanisms and interactions).

4. Figure 1 seems to be not too demonstrative, it is not clear, what is the difference between „individual differences (new line) context” and „individual. contextual (new line) differences”, it is also not clear, what is the role of full stop (.) in the 2nd form.

5. In general, the manuscript is full of oversophisticated terms, that probably makes it extreme hard to follow for not native English speakers, even for readers with advanced level of English proficiency.

Reviewer 3 Report

I enjoyed reading this perspective paper. The authors provide a great integration of past and current research regarding the social effects of oxytocin, connecting to compassion-related processes. The manuscript is well-written and structured. I suggest adding a discussion of oxytocin's effect in the context of chronic pain. That could be a great example proxy of suffering. Also, the authors may extend the storytelling with the citation of  Harari-Dahan & Bernstein (2014).

Round 2

Reviewer 2 Report

The manuscript has been improved during the revision process.

Author Response

It seems that the reviewer has been satisfied by the previous round of revisions:

"The manuscript has been improved during the revision process."